# Echocardiographic left ventricular stroke work index: An integrated noninvasive measure of shock severity

Jacob C. Jentzer[1,2]*, Brandon M. Wiley[1], Nandan S. Anavekar[1]

**1** Department of Cardiovascular Medicine, Mayo Clinic, Rochester, Minnesota, United States of America,
**2** Robert D. and Patricia E. Kern Center for the Science of Health Care Delivery, Mayo Clinic, Rochester, Minnesota, United States of America

* jentzer.jacob@mayo.edu

## Abstract

### Background

Echocardiographic findings vary with shock severity, as defined by the Society for Cardiovascular Angiography and Intervention (SCAI) shock stage. Left ventricular stroke work index (LVSWI) measured by transthoracic echocardiography (TTE) can predict mortality in the cardiac intensive care unit (CICU). We sought to determine whether LVSWI could refine mortality risk stratification by the SCAI shock classification in the CICU.

### Methods

We included consecutive CICU patients from 2007 to 2015 with TTE data available to calculate the LVSWI, specifically the mean arterial pressure, stroke volume index and medial mitral E/e' ratio. In-hospital mortality as a function of LVSWI was evaluated across the SCAI shock stages using logistic regression, before and after multivariable adjustment.

### Results

We included 3635 unique CICU patients, with a mean age of 68.1 ± 14.5 years (36.5% females); 61.1% of patients had an acute coronary syndrome. The LVSWI progressively decreased with increasing shock severity, as defined by increasing SCAI shock stage. A total of 203 (5.6%) patients died during hospitalization, with higher in-hospital mortality among patients with lower LVSWI (adjusted OR 0.66 per 10 J/m2 higher) or higher SCAI shock stage (adjusted OR 1.24 per each higher stage). A LVSWI <33 J/m2 was associated with higher adjusted in-hospital mortality, particularly among patients with shock (SCAI stages C, D and E).

### Conclusions

The LVSWI by TTE noninvasively characterizes the severity of shock, including both systolic and diastolic parameters, and can identify low-risk and high-risk patients at each level of clinical shock severity.

**Data Availability Statement:** All relevant data are within the paper and its Supporting Information files.

**Funding:** The author(s) received no specific funding for this work.

**Competing interests:** The authors have declared that no competing interests exist.

## Introduction

Cardiogenic shock is a leading cause of morbidity and mortality in the cardiac intensive care unit (CICU) [1–3]. The presentation of cardiogenic shock (CS) varies across a continuum of severity as defined by the Society for Cardiovascular Angiography and Intervention (SCAI) shock stages classification [4,5]. Clinical studies have consistently demonstrated an association between higher shock severity, as represented by increasing SCAI shock stage A (At risk) to E (Extremis), and higher mortality in patients with CS, as well as CICU patients [6–15]. Therefore, accurate classification of CS severity is imperative in order to guide therapeutic interventions and optimize clinical outcomes [5].

Impaired cardiac hemodynamics characterize the pathophysiology of CS, making accurate evaluation of these parameters fundamental to the clinical assessment of patients presenting with CS [16]. Abnormal cardiac hemodynamic indices, measured either invasively with a pulmonary artery catheter or noninvasively using Doppler echocardiography, are associated with mortality risk in CS patients and CICU patients [10,14,15,17–21]. While several derived hemodynamic parameters such as cardiac power output (CPO) have been proposed to predict outcomes and guide therapy in patients with CS, no established hemodynamic marker exists for quantifying the severity of myocardial impairment across the spectrum of shock severity [10,16,18–22].

The left ventricular stroke work index (LVSWI) is a beat-by-beat assessment of myocardial systolic and diastolic function that integrates systemic hemodynamics to produce a comprehensive measure of cardiac performance [17,23]. The LVSWI can be calculated using Doppler echocardiography (ECHO-LVSWI) based on the stroke volume index (SVI) and ratio of mitral valve E velocity to medial mitral annulus e' velocity (E/e' ratio, used to estimate left ventricular filling pressures), and has been found to be strongly associated with mortality in CICU patients [17,23]. In a prior analysis, we demonstrated that patients with a low SVI or high E/e' ratio had higher in-hospital mortality across the SCAI shock stages, making it likely that ECHO-LVSWI would be associated with mortality as well [10]. The ECHO-LVSWI appeared to decrease as the SCAI shock stage increased, suggesting a strong correlation with shock severity that might implicate ECHO-LVSWI as an integrated marker of hemodynamic compromise combining both systolic and diastolic left ventricular function [10].

Given the equipoise that exists regarding the ideal cardiac hemodynamics indices for defining CS severity, we hypothesized that ECHO-LVSWI, as an integrated measurement reflecting overall myocardial performance, may better characterize CS severity when evaluated early during the clinical course. Therefore, we sought to evaluate the association of ECHO-LVSWI with in-hospital mortality across SCAI shock stages and to determine whether early assessment of this hemodynamic variable could augment risk-stratification.

## Methods

### Study population

This study was approved by the Institutional Review Board of Mayo Clinic as posing minimal risk to patients and was performed under a waiver of informed consent. We retrospectively analyzed the index CICU admission of consecutive unique adult patients aged ≥18 years admitted to the CICU at Mayo Clinic Hospital St. Mary's Campus between January 1, 2007 and December 31, 2015 who had a transthoracic echocardiogram (TTE) performed within 1 day before or after CICU admission [3,6–11,24–28]. We excluded patients who did not have available data to calculate the ECHO-LVSWI (i.e. MAP, SVI and E/e' ratio).

## Data sources

We recorded demographic, vital sign, laboratory, clinical and outcome data, as well as procedures and therapies performed during the CICU and hospital stay [3,6–11,24–28]. The admission value of all vital signs, clinical measurements and laboratory values was defined as either the first value recorded after CICU admission or the value recorded closest to CICU admission. Admission diagnoses were defined as all International Classification of Diseases (ICD)-9 diagnostic codes within 1 day before or after CICU admission [3]. The Acute Physiology and Chronic Health Evaluation (APACHE)-III score, APACHE-IV predicted hospital mortality and Sequential Organ Failure Assessment score were automatically calculated with data from the first 24 hours of CICU admission using previously-validated electronic algorithms [3,24–26]. The Charlson Comorbidity Index and individual comorbidities were extracted from the medical record using a previously-validated electronic algorithm [3,6–11,17,24–28].

## Echocardiographic data

The Mayo Clinic Echocardiography Database was queried and data extracted from the TTE performed closest to CICU admission, including vital signs at the time of TTE (**S1 Table**) [10,17]. One best LVEF value for each patient was determined using a hierarchical approach: volumetric LVEF calculated using Simpson's biplane method was preferred, followed by monoplane volumetric approach, followed by linear methods and finally by visual estimation if these other methods were unavailable; the specific method used to measure the LVEF for each individual patient could not be determined [10,17]. We classified LVEF as mildly, moderately and severely reduced using gender-specific cut-offs as per current guidelines [29]. The mitral E/e' ratio was used to estimate left ventricular end-diastolic pressure as LVEDP = 4.9 + 0.62 * mitral E/e' ratio for calculation of ECHO-LVSWI using the formula ECHO-LVSWI = 0.0136 * stroke volume index (SVI) * (mean arterial pressure–LVEDP), as described by Choi, et al. (**S2 Table**) [17,23]. As per our prior study, we used the medial e' velocity to calculate ECHO-LVSWI (**Fig 1**); ECHO-LVSWI values were very similar when using either the lateral e' velocity or mean e' velocity (Pearson r correlation coefficients >0.99) [17]. Among patients with data for both medial and septal e' velocities, there were no significant differences between the AUC values for discrimination of in-hospital mortality with ECHO-LVSWI calculated using the medial (0.756), lateral (0.751) or mean (0.754) e' velocities

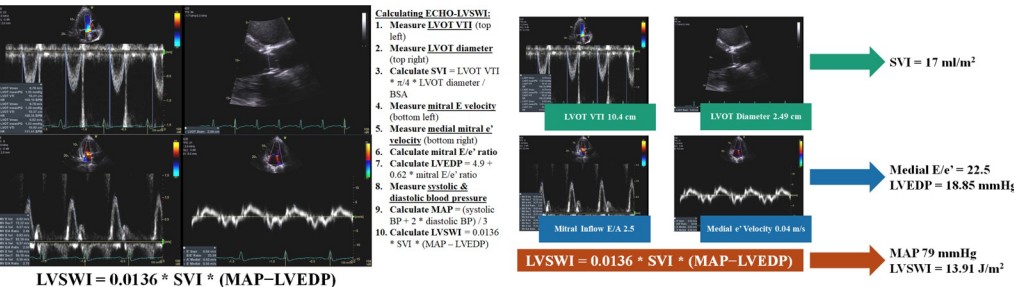

**Fig 1.** Calculation of the ECHO-LVSWI (left) and an example using TTE data from a patient with cardiogenic shock (right). The stroke volume index (SVI) is calculated using the left ventricular outflow tract (LVOT) velocity-time integral (VTI) by spectral Doppler, indexed to the body surface area. The LVEDP is estimated using the ratio of the peak mitral early diastolic (E) wave velocity by spectral Doppler to the peak mitral early diastolic (e') wave velocity by tissue Doppler via the formula LVEDP = 4.9 + 0.62 * mitral E/e' ratio [17,23]. We used the medial/septal mitral e' velocity for this analysis, although our data suggest that either the lateral or mean e' velocity could be substituted. The mean arterial pressure (MAP) was determined either invasively or noninvasively and estimated as (systolic blood pressure + 2 * diastolic blood pressure) / 3. We used the formula ECHO-LVSWI = 0.0136 * SVI * (MAP–LVEDP) [17,23].

(all p >0.05 by De Long test; **S1 Fig**). Per Mayo Clinic Echocardiography Laboratory policy, mitral E velocities were not reported for patients with E-A fusion. Right atrial pressure was either recorded at the time of TTE (if measured invasively) or estimated based on inferior vena cava size and collapsibility [30].

## Definition of SCAI shock stages

We defined hemodynamic instability (including the need for inotropes), hypoperfusion (including the need for vasopressors), deterioration and refractory shock using data from CICU admission through the first 24 hours in the CICU (**S3 Table**) [6–11]. We mapped the five SCAI shock stages with increasing severity (A through E) using combinations of these variables, using an algorithm based on our prior analyses (**S4 Table**) [4,6–11]. Due to the small number of included patients in SCAI shock stage E, we grouped patients with SCAI shock stages D and E together for this analysis [7,10].

## Statistical analysis

In-hospital mortality was determined using electronic review of health records. Variables of interest were compared across the SCAI shock stages, and relevant analyses repeated in each SCAI shock stage. Categorical variables are reported as number (percentage) and the Pearson chi-squared test was used to compare groups; trends across the SCAI shock stages were determined using logistic regression. Continuous variables are reported as mean ± standard deviation and Student's t test was used to compare groups; trends across the SCAI shock stages were determined using linear regression. Classification and regression tree (CART) analysis was used to identify 4 risk groups using ECHO-LVSWI and SCAI shock stage. Discrimination of in-hospital mortality was assessed using area under the receiver-operator characteristic curve (AUC) values, which were compared using the De Long test. Logistic regression was used to determine odds ratio (OR) and 95% confidence interval (CI) values for prediction of in-hospital mortality, before and after multivariable adjustment. Multivariable regression was performed using stepwise backward variable selection to minimize the value of the Akaike Information Criterion (AIC, a measure reflecting deviation from ideal model performance in the population). ECHO-LVSWI was analyzed as a continuous variable, dichotomized by the optimal cut-off, and according to prespecified categories. Candidate variables included demographics, comorbidities, admission diagnoses, severity of illness scores (including SCAI shock stage), LVEF and procedures and therapies. Statistical analyses were performed using JMP Pro version 14.1.0 (SAS Institute, Cary, NC).

## Results

### Study population

Out of a database of 10,004 unique CICU patients, we excluded 6,369 patients: 317 patients without an echocardiogram, 1,299 whose echocardiogram was not a TTE, 2,482 patients whose TTE was more than one day before or after CICU admission, and 2,271 patients whose TTE did not have data available to calculate the ECHO-LVSWI (**Fig 2**). The remaining 3,635 patients comprising the final study population had a mean age of 68.1 ± 14.5 years (36.5% females). Admission diagnoses included acute coronary syndrome in 61.1%, heart failure in 43.6%, cardiac arrest in 11.3%, cardiogenic shock in 9.4% and sepsis in 4.9%. The distribution of SCAI shock stages was: A, 50.8%; B, 27.9%; C, 15.6%; D, 5.3%; E, 0.4%.

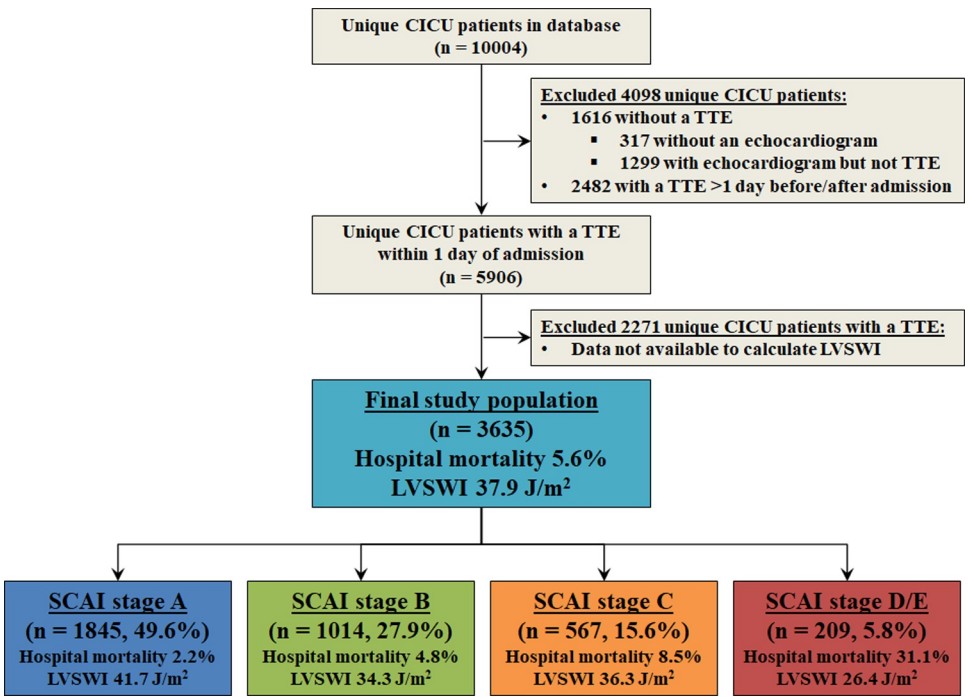

**Fig 2. Flow diagram showing study inclusion and exclusion criteria.** CICU, cardiac intensive care unit; LVSWI, left ventricular stroke work index; SCAI, Society for cardiovascular Angiography and Intervention; TTE, transthoracic echocardiogram.

## Echocardiographic findings

TTE was performed on the day of CICU admission in 42.9%. The mean LVEF was 48.3 ± 15.6%, and 51.8% patients had at least mildly reduced LVEF. The mean SVI was 40.7 ± 10.9 ml/m2, the mean E/e' ratio was 15.9 ± 9.1, and the mean MAP at the time of the TTE was 82.9 ± 14.0 mmHg. The mean ECHO-LVSWI was 37.9 ± 13.8 J/m2, with a distribution as follows (in J/m2): ≥50, 17.7%; 40–49, 23.2%; 30–39, 30.3%; 20–29, 20.1%; <20, 8.6%. Baseline characteristics varied substantially as a function of SCAI shock stage (**Table 1**), as did echocardiographic findings (**Table 2**). The ECHO-LVSWI decreased with increasing SCAI shock stage (**Fig 3**), with a distribution shifted toward lower values of ECHO-LVSWI (**Fig 3**); no patient in SCAI shock stage E had a LVSWI ≥40 J/m2.

## In-hospital mortality–unadjusted analyses

A total of 203 (5.6%) patients died during hospitalization. The ECHO-LVSWI was lower among inpatient deaths compared to hospital survivors (27.0 versus 38.6 J/m2, p <0.001). ECHO-LVSWI was strongly and inversely associated with in-hospital mortality (unadjusted OR 0.453 per 10 J/m2 higher ECHO-LVSWI, 95% CI 0.396–0.518, p <0.001). The optimal ECHO-LVSWI cut-off for prediction of in-hospital death was 33.0 J/m2, and patients with ECHO-LVSWI <33.0 J/m2 had higher in-hospital mortality (10.9% versus 2.3%, unadjusted OR 5.112, 95% CI 3.709–7.046, p <0.001), accounting for 73.9% of in-hospital deaths. The association between ECHO-LVSWI and in-hospital mortality was weaker for patients with a heart rate >90 beats/minute (unadjusted OR 0.585 per 10 J/m2 higher ECHO-LVSWI, 95% CI 0.442–0.776, AUC 0.645) compared with patients who had a slower heart rate (unadjusted OR 0.461 per 10 J/m2 higher ECHO-LVSWI, 95% CI 0.392–0.543, AUC 0.740). LVSWI had a

**Table 1. Baseline characteristics, comorbidities, admission diagnoses and therapies of patients according to SCAI shock stages.** Data reported as mean ± standard deviation for continuous variables and number (percent) for categorical variables. P value is for linear regression (continuous variables) or logistic regression (categorical variables) across SCAI shock stages.

| Variable | Stage A (n = 1845) | Stage B (n = 1014) | Stage C (n = 567) | Stage D/E (n = 209) | P value |
|---|---|---|---|---|---|
| *Demographics and outcomes* | | | | | |
| **Age** | 67.6±14.2 | 67.3±15.0 | 71.2±14.4 | 68.7±14.1 | <0.001 |
| **Female gender** | 579 (31.4%) | 418 (41.2%) | 234 (41.3%) | 73 (34.9%) | <0.001 |
| **White race** | 1722 (93.3%) | 939 (92.6%) | 520 (91.7%) | 192 (91.9%) | 0.16 |
| **CICU length of stay** | 2.1±5.6 | 2.5±2.4 | 2.5±2.3 | 5.0±4.2 | <0.001 |
| **Hospital length of stay** | 5.4±8.8 | 7.0±7.5 | 6.9±7.6 | 11.5±11.6 | <0.001 |
| **CICU mortality** | 20 (1.1%) | 27 (2.7%) | 30 (5.3%) | 54 (25.8%) | <0.001 |
| **Hospital mortality** | 41 (2.2%) | 49 (4.8%) | 48 (8.5%) | 65 (31.1%) | <0.001 |
| *Comorbidities* | | | | | |
| **Charlson Comorbidity Index** | 1.9±2.4 | 2.2±2.5 | 2.6±2.7 | 2.9±2.8 | <0.001 |
| **History of MI** | 356 (19.3%) | 197 (19.5%) | 120 (21.2%) | 43 (20.6%) | 0.38 |
| **History of HF** | 207 (11.2%) | 157 (15.5%) | 88 (15.5%) | 55 (26.3%) | <0.001 |
| **History of CKD** | 254 (13.8%) | 166 (16.4%) | 118 (20.8%) | 55 (26.3%) | <0.001 |
| **Prior dialysis** | 41 (2.2%) | 40 (3.9%) | 52 (9.2%) | 29 (13.9%) | <0.001 |
| *Admission diagnoses* | | | | | |
| **ACS** | 1202 (66.2%) | 564 (56.0%) | 325 (58.2%) | 103 (49.8%) | <0.001 |
| **HF** | 629 (34.6%) | 511 (50.7%) | 273 (48.9%) | 153 (73.9%) | <0.001 |
| **Cardiac arrest** | 132 (7.3%) | 110 (10.9) | 80 (14.3%) | 83 (40.1%) | <0.001 |
| **VF arrest** | 85 (4.7%) | 69 (6.8%) | 40 (7.2%) | 41 (19.8%) | <0.001 |
| **Respiratory failure** | 191 (10.5%) | 219 (21.8%) | 114 (20.4%) | 141 (68.1%) | <0.001 |
| **Sepsis** | 32 (1.8%) | 63 (6.3%) | 35 (6.3%) | 47 (22.7%) | <0.001 |
| *Therapies and procedures* | | | | | |
| **Any vasoactive infusions** | 164 (8.9%) | 213 (21.0%) | 55 (9.7%) | 195 (93.3%) | <0.001 |
| **Vasopressors** | 139 (7.5%) | 192 (18.9%) | 48 (8.5%) | 181 (86.6%) | <0.001 |
| **Inotropes** | 52 (2.8%) | 61 (6.0%) | 13 (2.3%) | 62 (29.7%) | <0.001 |
| **# vasoactives** | 0.1±0.5 | 0.3±0.8 | 0.1±0.5 | 2.0±1.2 | <0.001 |
| **Peak VIS** | 1.0±7.3 | 3.5±12.9 | 1.5±8.1 | 34.9±58.8 | <0.001 |
| **Peak NEE (mcg/kg/min)** | 0.01±0.08 | 0.03±0.13 | 0.01±0.08 | 0.33±0.59 | <0.001 |
| **Invasive ventilator** | 99 (5.4%) | 138 (13.6%) | 90 (15.9%) | 135 (64.6%) | <0.001 |
| **Noninvasive ventilator** | 187 (10.1%) | 171 (16.9%) | 85 (15.0%) | 66 (31.6%) | <0.001 |
| **Dialysis in CICU** | 26 (1.4%) | 32 (3.2%) | 16 (2.8%) | 33 (15.8%) | <0.001 |
| **CRRT** | 1 (0.1%) | 8 (0.8%) | 5 (0.9%) | 24 (11.5%) | <0.001 |
| **IABP in CICU** | 98 (5.3%) | 116 (11.4%) | 30 (5.3%) | 57 (27.3%) | <0.001 |
| **PAC in CICU** | 38 (2.1%) | 57 (5.6%) | 12 (2.1%) | 51 (24.4%) | <0.001 |
| **Coronary angiogram** | 1333 (72.2%) | 632 (62.3%) | 337 (59.4%) | 117 (56.0%) | <0.001 |
| **PCI** | 955 (51.8%) | 409 (40.3%) | 194 (34.2%) | 61 (29.2%) | <0.001 |
| **RBC transfusion** | 102 (5.5%) | 110 (10.8%) | 60 (10.6%) | 70 (33.5%) | <0.001 |
| **In-hospital CPR** | 22 (1.2%) | 21 (2.1%) | 12 (2.1%) | 27 (12.9%) | <0.001 |
| *Severity of illness* | | | | | |
| **APACHE-III score** | 50.4±17.7 | 59.8±20.2 | 66.8±23.1 | 96.0±31.0 | <0.001 |
| **APACHE-IV predicted death (%)** | 9.1±10.5 | 15.0±15.8 | 19.8±19.6 | 46.5±28.9 | <0.001 |
| **Day 1 SOFA score** | 2.1±1.8 | 3.1±2.6 | 3.9±2.7 | 9.3±3.9 | <0.001 |
| **Non-cardiovascular SOFA** | 1.1±1.6 | 1.9±2.3 | 2.9±2.6 | 6.5±3.3 | <0.001 |
| **Non-cardiovascular organ failure** | 201 (10.9%) | 212 (20.9%) | 252 (44.4%) | 168 (80.4%) | <0.001 |
| **SIRS on admission** | 293 (15.9%) | 430 (42.4%) | 231 (40.7%) | 139 (66.5%) | <0.001 |
| **Admission Braden score** | 18.6±2.8 | 17.6±3.2 | 17.4±3.3 | 14.3±3.5 | <0.001 |

*(Continued)*

**Table 1.** (Continued)

| Variable | Stage A (n = 1845) | Stage B (n = 1014) | Stage C (n = 567) | Stage D/E (n = 209) | P value |
|---|---|---|---|---|---|
| CardShock score | 1.6±1.1 | 1.7±1.2 | 2.5±1.5 | 3.5±1.7 | <0.001 |

Abbreviations: ACS, acute coronary syndrome; APACHE, Acute Physiology and Chronic Health Evaluation; CICU, cardiac intensive care unit; CKD, chronic kidney disease; CPR, cardiopulmonary resuscitation; HF, heart failure; IABP, intra-aortic balloon pump; MI, myocardial infarction; NEE, norepinephrine-equivalent dose; PAC, pulmonary artery catheter; PCI, percutaneous coronary intervention; RBC, red blood cell; SOFA, Sequential Organ Failure Assessment; VIS, Vasoactive-Inotropic Score.

* Admission diagnoses are not mutually-exclusive and sum to greater than 100%.

higher AUC value (S2 Fig) than LVEF (AUC 0.662, p <0.001), cardiac index (AUC 0.604, p <0.001) and SVI (AUC 0.702, p = 0.06). When patients were grouped by quartiles of these variables, ECHO-LVSWI produced the greatest separation between high-risk and low-risk patients (S3 and S4 Figs and Table 3). At the optimal cut-off, LVSWI had the highest combined sensitivity and specificity (Table 4).

In-hospital mortality increased with lower ECHO-LVSWI and higher SCAI shock stage (Fig 4), and patients with ECHO-LVSWI <33.0 J/m2 had higher in-hospital mortality in each SCAI shock stage (all p <0.05; Fig 4). Patients in each lower SCAI shock stage who had ECHO-LVSWI <33.0 J/m2 had similar in-hospital mortality as patients in the next higher SCAI shock stage with ECHO-LVSWI $\geq$33.0 J/m$^2$ (all p >0.1). Patients in SCAI shock stage D/E with ECHO-LVSWI <33.0 J/m$^2$ had the highest in-hospital mortality. ECHO-LVSWI was inversely associated with in-hospital mortality risk in each SCAI shock stage (Fig 5, all p <0.01). Echo-LVSWI alone had an AUC of 0.747 for discrimination of in-hospital mortality, which was equivalent to that for the SCAI shock stages (0.753, p = 0.78 by De Long test). The combination of ECHO-LVSWI and SCAI shock stage had an AUC of 0.803 for discrimination of in-hospital mortality, which was higher than either ECHO-LVSWI or SCAI shock stage alone (p <0.001). CART analysis separated patients into 4 risk groups based on ECHO-LVSWI (using the cut-off of 33.1 J/m$^2$) and SCAI shock stage A/B/C versus D/E (Fig 6). In-hospital mortality increased from 2.0% among patients in SCAI shock stage A/B/C with ECHO-LVSWI $\geq$33.1 J/m$^2$ to 35.0% among patients in SCAI shock stage D/E with ECHO-LVSWI <33.1 J/m$^2$ (Fig 6).

### In-hospital mortality–multivariable analysis

The final multivariable model had an AUC of 0.93 for discrimination of in-hospital mortality (Table 5). After adjustment, ECHO-LVSWI remained strongly and inversely associated with in-hospital mortality (adjusted OR 0.664 per 10 J/m2 higher ECHO-LVSWI, 95% CI 0.564–0.782, p <0.001); ECHO-LVSWI had the second highest log worth in the model, after cardiac arrest. Patients with ECHO-LVSWI <33.0 J/m2 had higher adjusted in-hospital mortality (adjusted OR 2.232, 95% CI 1.507–3.305, p <0.001). When compared with patients who had ECHO-LVSWI <20 J/m2, patients in each higher ECHO-LVSWI group had lower adjusted in-hospital mortality (all p <0.05). When compared with patients who had ECHO-LVSWI $\geq$50 J/m2, patients in each lower ECHO-LVSWI group had higher adjusted in-hospital mortality (all p <0.05). When multivariable regression was repeated separately for each SCAI shock stage, a higher ECHO-LVSWI was associated with lower in-hospital mortality in each SCAI shock stage (all p <0.05 except SCAI shock stage B, p = 0.13; Fig 5). Patients with ECHO-LVSWI <33.0 J/m2 had higher adjusted in-hospital mortality in SCAI shock stage C (p <0.001) and D/E (p <0.05), but not in SCAI shock stage A (p = 0.17) or B (p = 0.81).

**Table 2. Echocardiographic findings of patients according to SCAI shock stages.** Data reported as mean ± standard deviation for continuous variables and number (percent) for categorical variables. P value is for linear regression (continuous variables) or logistic regression (categorical variables) across SCAI shock stages.

| Variable | n with data | Stage A (n = 1845) | Stage B (n = 1014) | Stage C (n = 567) | Stage D/E (n = 209) | P value |
|---|---|---|---|---|---|---|
| *Vital signs at TTE* | | | | | | |
| **TTE on day of admission** | 3635 | 780 (42.3%) | 440 (43.4%) | 241 (42.5%) | 98 (46.9%) | 0.34 |
| **Systolic BP (mmHg)** | 3635 | 122.5±20.3 | 114.2±20.8 | 117.6±21.1 | 105.0±19.6 | <0.001 |
| **Diastolic BP (mmHg)** | 3635 | 67.1±12.8 | 64.5±14.6 | 63.9±13.8 | 59.0±13.4 | <0.001 |
| **Mean BP (mmHg)** | 3635 | 85.6±13.0 | 81.0±14.6 | 81.8±13.8 | 74.3±13.5 | <0.001 |
| **Pulse pressure (mmHg)** | 3635 | 55.4±18.8 | 49.8±18.2 | 53.7±19.6 | 46.0±17.2 | <0.001 |
| **Heart rate (BPM)** | 3519 | 68.8±12.9 | 79.2±18.6 | 75 (16.6%) | 79.9±19.0 | <0.001 |
| **Shock index** | 3519 | 0.58±0.15 | 0.72±0.22 | 0.66±0.21 | 0.79±0.26 | <0.001 |
| **Atrial fibrillation** | 3478 | 114 (6.5%) | 159 (17.8%) | 83 (14.3%) | 48 (18.5%) | <0.001 |
| *LV systolic function* | | | | | | |
| **LVEDD** | 3492 | 51.2±7.2 | 51.7±8.0 | 50.9±7.9 | 52.7±10.1 | 0.09 |
| **LVESD** | 3023 | 35.9±8.7 | 38.3±10.6 | 36.8±10.1 | 41.0±12.7 | <0.001 |
| **Fractional shortening (%)** | 3019 | 30.1±8.9 | 26.9±10.5 | 28.2±10.5 | 23.5±11.2 | <0.001 |
| **LVEF (%)** | 3608 | 50.8±13.9 | 46.4±16.4 | 47.4±16.3 | 40.0±17.2 | <0.001 |
| **LVSD by ASE criteria** | 3608 | 810 (45.2%) | 531 (56.4%) | 322 (54.0%) | 206 (74.1%) | <0.001 |
| Mild LVSD | | 356 (19.9%) | 186 (19.8%) | 112 (18.8%) | 54 (19.4%) | |
| Moderate LVSD | | 303 (16.9%) | 174 (18.5%) | 111 (18.6%) | 71 (25.5%) | |
| Severe LVSD | | 151 (8.4%) | 171 (18.2%) | 99 (16.6%) | 81 (29.1%) | |
| **Wall motion score index** | 2432 | 1.7±0.4 | 1.8±0.5 | 1.8±0.5 | 2.0±0.5 | <0.001 |
| **Lateral mitral s' (cm/s)** | 2676 | 7.5±2.3 | 7.3±2.6 | 7.2±2.5 | 7.2±3.1 | 0.02 |
| *Systemic hemodynamics* | | | | | | |
| **LVOT peak velocity (m/s)** | 3632 | 1.0±0.2 | 1.0±0.2 | 1.0±02 | 1.0±0.2 | <0.001 |
| **LVOT VTI (cm)** | 3635 | 21.2±4.5 | 19.3±5.1 | 19. 6±5.0 | 17.0±4.9 | <0.001 |
| **SV (ml)** | 3635 | 85.4±21.4 | 76.0±23.3 | 76.3±23.0 | 66.6±22.3 | <0.001 |
| **SVI (ml/m²)** | 3635 | 43.1±9.8 | 38.9±11.5 | 39.7±11.1 | 34.0±11.1 | <0.001 |
| **LVSW (g*min)** | 3635 | 83.6±28.4 | 68.4±26.7 | 69.3±27.6 | 52.6±23.4 | <0.001 |
| **LVSWI (g*min/m²)** | 3635 | 42.0±13.2 | 34.8±12.8 | 35.9±13.3 | 26.7±11.4 | <0.001 |
| **LVSWI <33 g*min/m²** | 3635 | 464 (25.2%) | 517 (51.0%) | 229 (40.4%) | 163 (78.0%) | <0.001 |
| **MCF** | 3201 | 0.45±0.15 | 0.40±0.16 | 0.41±0.16 | 0.34±0.14 | <0.001 |
| **CO (L/min)** | 3602 | 5.7±1.4 | 5.8±1.7 | 5.5±1.7 | 5.1±1.8 | <0.001 |
| **CI (L/min/m²)** | 3602 | 2.9±0.7 | 3.0±0.8 | 2.9±0.8 | 2.6±0.9 | <0.001 |
| **CPO (W)** | 3602 | 1.1±0.3 | 1.0±0.4 | 1.0±0.4 | 0.8±0.3 | <0.001 |
| **CPI (W/m²)** | 3602 | 0.5±0.2 | 0.5±0.2 | 0.5±0.2 | 0.4±0.2 | <0.001 |
| **SVR (dyne*s/cm⁵)** | 3275 | 1157±350 | 1072±568 | 1143±421 | 1107±476 | 0.04 |
| **SVR index (dyne*s/cm⁵*m²)** | 3275 | 2260±657 | 2065±1122 | 2163±756 | 2158±940 | 0.001 |
| *LV diastolic function* | | | | | | |
| **Mitral E velocity (cm/s)** | 3635 | 0.8±0.3 | 0.9±0.3 | 0.9±0.3 | 0.8±0.3 | <0.001 |
| **Mitral E/A ratio** | 3031 | 1.1±0.6 | 1.3±0.7 | 1.2±0.7 | 1.3±0.8 | <0.001 |
| **Mitral e' velocity (cm/s)** | 3635 | 6.1±2.2 | 6.2±2.4 | 5.7±2.3 | 5.1±2.1 | <0.001 |
| **Mitral E/e' ratio** | 3635 | 14.9±8.3 | 16.1±9.2 | 17.3±9.6 | 18.8±11.0 | <0.001 |
| **Mitral E DT (ms)** | 3207 | 202.1±54.8 | 182.2±51.4 | 192.6±54.6 | 177,9±51.9 | <0.001 |
| *RV function* | | | | | | |
| **Estimated RAP (mmHg)** | 3302 | 7.8±4.2 | 9.7±5.0 | 9.8±5.0 | 13.1±5.3 | <0.001 |
| **Pressure-adjusted heart rate** | 3198 | 6.5±4.2 | 10.0±6.7 | 9.5±6.1 | 14.3±7.2 | <0.001 |
| **Peak TR velocity (m/s)** | 2896 | 2.7±0.5 | 2.7±0.5 | 2.8±0.5 | 2.7±0.6 | 0.52 |
| **Estimated RVSP (mmHg)** | 2876 | 38.5±13.8 | 50.9±13.7 | 41.4±13.0 | 43.7±14.2 | <0.001 |

*(Continued)*

**Table 2.** (Continued)

| Variable | n with data | Stage A (n = 1845) | Stage B (n = 1014) | Stage C (n = 567) | Stage D/E (n = 209) | P value |
|---|---|---|---|---|---|---|
| **Tricuspid s' (cm/s)** | 2848 | 11.9±3.2 | 11.4±3.5 | 11.6±3.5 | 10.2±4.2 | <0.001 |
| **Global RV dysfunction** | 2073 | 401 (41.3%) | 305 (58.1%) | 214 (59.4%) | 169 (78.2%) | <0.001 |
| **Mild/mild-moderate** | | 245 (25.2%) | 160 (30.5%) | 110 (30.6%) | 68 (31.5%) | |
| **Moderate/severe** | | 156 (16.05) | 145 (27.6%) | 104 (28.9%) | 101 (46.8%) | |

Abbreviations: BP, blood pressure; CO, cardiac output; CI, cardiac index, CPO, cardiac power output; DT, deceleration time; LVEF, left ventricular ejection fraction; LVOT, left ventricular outflow tract; LVSW, left ventricular stroke work; LVSWI, left ventricular stroke work index; MCF, myocardial contraction fraction; RAP, right atrial pressure; RVSP, right ventricular systolic pressure; SV, stroke volume; SVI, stroke volume index; SVR, systemic vascular resistance; TR, tricuspid regurgitation; TTE, transthoracic echocardiogram; VTI, velocity-time integral.

## Discussion

The ECHO-LVSWI is a noninvasive measure of cardiac function that integrates relevant systolic, diastolic and systemic parameters to quantify the degree of hemodynamic compromise in CICU patients with or at risk for CS. The ECHO-LVSWI assessed early in the clinical course is a powerful echocardiographic predictor of in-hospital mortality among CICU patients, even after adjusting for standard measures of shock severity and overall illness severity. The ECHO-LVSWI was inversely associated with in-hospital mortality risk, and patients at the extremes of ECHO-LVSWI ($<20$ J/m2 and $\geq50$ J/m2) had substantially different mortality than patients with intermediate values (with an optimal cut-off ~33 J/m2).

When evaluated with respect to the SCAI stages of shock, ECHO-LVSWI demonstrated clear incremental value for risk stratification. As expected, ECHO-LVSWI valves progressively declined as the severity of shock worsened from SCAI shock stage C to D to E, reflecting worsening myocardial performance and systemic hemodynamics. Calculation of LVSWI using the mean invasive hemodynamic values reported by Thayer, et al. likewise demonstrates a drop in LVSWI as SCAI shock stage increases, supporting the validity of this finding [14]. What is more notable is that our analysis demonstrated that within each SCAI shock stage, decreasing ECHO-LVSWI values were associated with higher in-hospital mortality. Patients clinically classified into less severe SCAI shock stages who had paradoxically low ECHO-LVSWI had in-hospital mortality similar to patients in the next higher SCAI shock stage who had preserved ECHO-LVSWI. This finding suggests that ECHO-LVSWI can reclassify patients into higher-

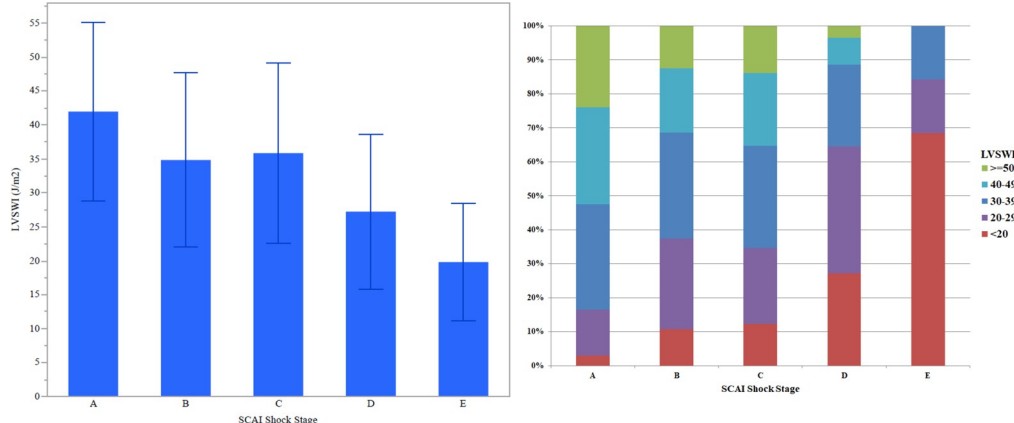

**Fig 3.** Mean ECHO-LVSWI (left) and distribution of ECHO-LVSWI (right) as a function of SCAI shock stage.

**Table 3. Unadjusted odds ratio (OR) and 95% confidence interval values for quartiles of selected echocardiographic for prediction of in-hospital mortality using univariable logistic regression, with Quartile 4 as referent.** Median and interquartile range values defining the quartiles are as follows: ECHO LVSWI, 37.0 (21.0, 46.1) J/m2; CI, 2.8 (2.4, 3.3) L/min/m2; LVEF, 51 (36, 61) %; SVI, 41 (33, 47) ml/m2.

| Quartile | LVEF | CI | SVI | LVSWI |
|---|---|---|---|---|
| 1 | 3.91 (2.52–6.05) | 2.62 (1.74–3.95) | 7.21 (4.31–12.05) | 15.83 (7.68–32.64) |
| 2 | 1.99 (1.23–3.20) | 1.12 (0.70–1.80) | 3.48 (2.04–5.93) | 6.97 (3.29–14.74) |
| 3 | 1.17 (0.69–1.98) | 1.25 (0.79–1.98) | 2.04 (1.13–3.68) | 3.84 (1.75–8.41) |
| 4 (referent) | 1.0 | 1.0 | 1.0 | 1.0 |

risk and lower-risk subgroups within the SCAI shock stage schema. ECHO-LVSWI permits the identification of clinically relevant myocardial dysfunction that might escape detection by the clinical exam or basic echocardiography. Performed early in the CICU stay, ECHO-LVSWI functioned as well as the SCAI shock stages schema for discrimination of in-hospital mortality, and incrementally improved mortality risk-stratification by the SCAI shock stages. The incremental risk-stratification provided by ECHO-LVSWI on top of the SCAI shock stages strengthens the argument that noninvasive hemodynamics should be routinely incorporated into shock severity assessment. The performance of a limited transthoracic echocardiogram or point-of-care ultrasound without the integration of Doppler derived hemodynamics may be inadequate for optimal risk stratification for CICU patients with or at risk of CS.

This analysis must be contrasted with our recent studies to highlight its incremental value [10,17]. We previously examined several other echocardiographic variables in CICU patients across the SCAI shock stages, finding that a SVI <35 ml/m2 and a medial mitral E/e' ratio >15 were independently associated with higher in-hospital mortality, while LVEF <40%, cardiac index <1.8 L/min/m2 and cardiac power output (CPO) were not [10]. Insofar as the ECHO-LVSWI is calculated using the SVI and mitral E/e' ratio, the results of the present study demonstrating the predictive value of ECHO-LVSWI are not surprising. We have demonstrated that ECHO-LVSWI has higher discrimination for in-hospital mortality than LVEF, SVI and CI when analyzed as continuous variables, providing risk stratification by separating high-risk and low-risk CICU patients. While our prior study did demonstrate that ECHO-LVSWI decreased across the SCAI shock stages, the present study is the first to examine the association between ECHO-LVSWI and outcomes as a function of shock severity [10]. In a larger analysis of unselected CICU patients, we previously reported the strong inverse association between ECHO-LVSWI and in-hospital mortality; however, this prior study did not account for SCAI shock stage [17].

By integrating the concepts demonstrated in these prior analyses, this study expands on the utility of echocardiographic ECHO-LVSWI for mortality risk-stratification in CICU patients across the spectrum of shock severity. Indeed, the ECHO-LVSWI has the highest

**Table 4. Optimal cut-off (maximum value of Youden's J index = sensitivity + specificity– 1) with the associated sensitivity, specificity and overall accuracy for selected echocardiographic variables for prediction of in-hospital mortality using univariable logistic regression.**

| Quartile | LVEF (%) | CI (L/min/m2) | SVI (ml/m2) | LVSWI (J/m2) |
|---|---|---|---|---|
| Optimal cut-off | 45% | 2.50 L/min/m2 | 36 ml/m2 | 33.0 |
| Sensitivity | 65.7% | 48.2% | 66.5% | 73.9% |
| Specificity | 60.6% | 71.4% | 67.2% | 64.2% |
| Overall accuracy | 60.9% | 70.1% | 67.2% | 64.7% |

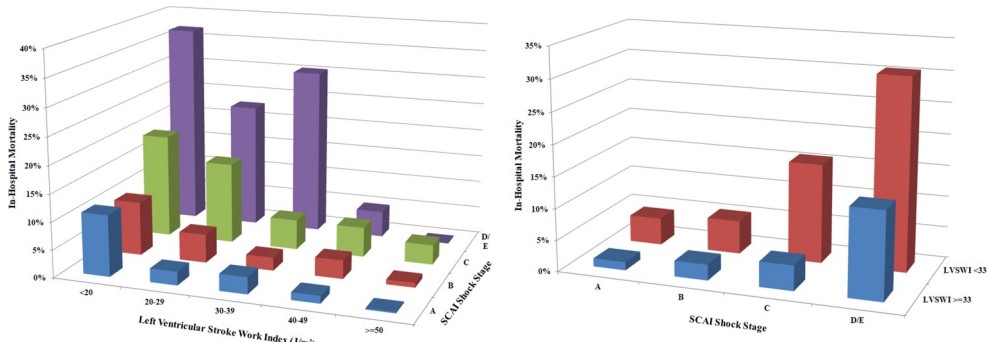

**Fig 4.** In-hospital mortality as a function of SCAI shock stage stratified by ECHO-LVSWI group (left) and for patients with ECHO-LVSWI < or ≥33.1 J/m² (right).

discrimination value for in-hospital mortality of any of the echocardiographic variables we have examined, and was a more important predictor of in-hospital mortality than SCAI shock stage on multivariable regression and CART analysis [10,17]. In-hospital mortality was low among patients with preserved ECHO-LVSWI, even in the presence of severe (SCAI stage D/E) shock. Likewise, patients with low ECHO-LVSWI (especially <20 J/m2) had high in-hospital mortality, even in the absence of hemodynamic instability or shock during the first 24 hours after CICU admission. Therefore, the ECHO-LVSWI provided incremental refinement of prognosis at each SCAI shock stage, allowing identification of higher-risk and lower-risk subgroups that might require different approaches to management. The measurement of ECHO-LVSWI by TTE may be used to enhance prognostication and facilitate care in CICU patients by assessing the underlying degree of hemodynamic compromise beyond clinical assessment alone.

The search for an optimal variable to define cardiac performance and hemodynamic compromise in CS is ongoing, with several candidate variables identified. Several authors have proposed the CPO, derived from the MAP and cardiac output, as the preferred hemodynamic parameter for both prognosticating and guiding clinical care in CS patients [18–22]. This is logical as the MAP and cardiac output are relevant determinants of organ perfusion which can be manipulated with therapeutic interventions. However, when measured noninvasively using echocardiography, CPO is not as strongly associated with mortality as beat-by-beat parameters

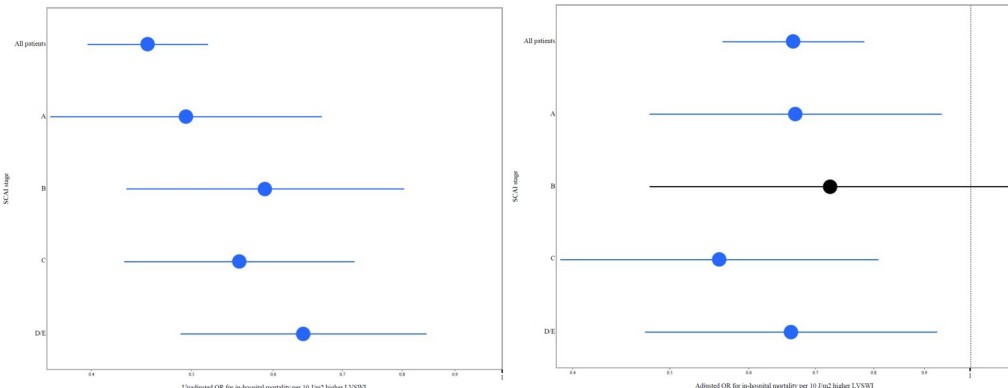

**Fig 5.** Forest plots showing unadjusted (left) and adjusted (right) odds ratio (OR) values for ECHO-LVSWI (per each 10 J/m² higher) overall and in each SCAI shock stage.

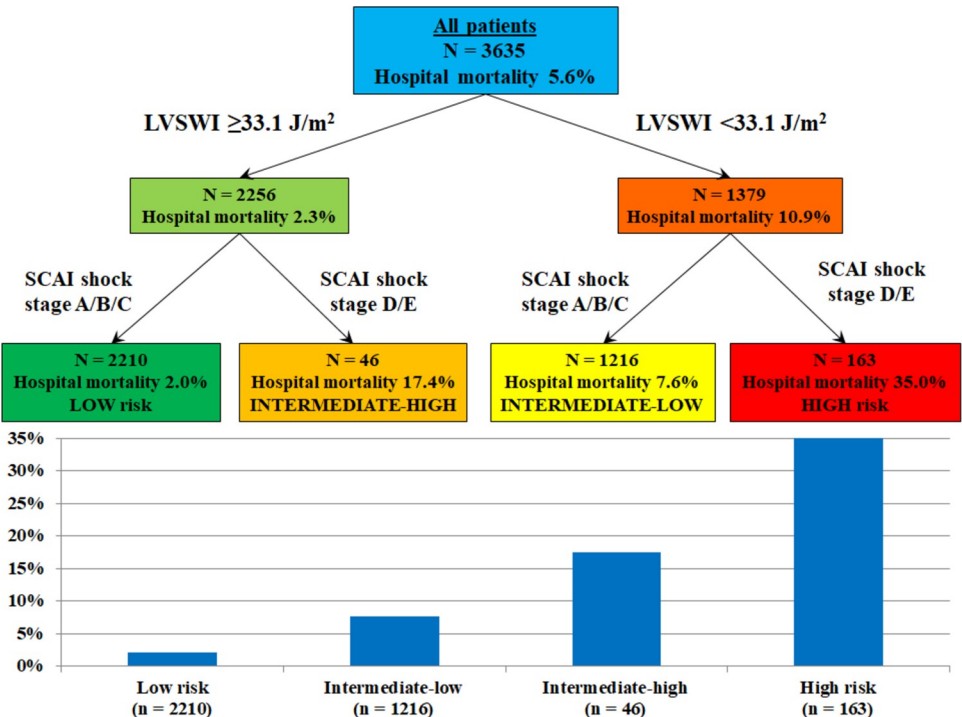

**Fig 6. Classification and regression tree (CART) analysis using ECHO-LVSWI and SCAI shock stage for stratification of in-hospital mortality risk.**

such as SVI or ECHO-LVSWI; this could relate to the confounding effect of heart rate, which may compensate for low SV in some patients with circulatory failure and contribute to measurement error [10,17,31]. Recent multicenter analyses did not demonstrate substantial variation in invasively-measured CI or CPO across SCAI shock stage, and neither CI nor CPO was associated with mortality [14,15]. This suggests that, while it remains a logical target for titrating therapy in CS patients, CPO may not be the ideal hemodynamic parameter for predicting outcomes in this population.

ECHO-LVSWI may prove to be a superior non-invasive parameter because it integrates diastolic function assessment, providing deeper insights into overall myocardial function that translates into improved mortality risk stratification. Another echocardiographic measure of cardiac performance is the myocardial contraction fraction (MCF), which indexes the SV to the myocardial volume to quantify how efficiently the myocardium is pumping [32]. MCF has theoretical advantages over the LVEF as a measure of LV systolic function, and like ECHO-LVSWI the MCF uses forward SV to quantify cardiac function. No prior studies have examined MCF in the context of shock severity, and expectedly we observed a decrease in the MCF as SCAI shock stage increased. ECHO-LVSWI was a stronger echocardiographic predictor of in-hospital mortality than MCF in this cohort, presumably resulting from its added prognostic value resulting from inclusion of diastolic function (represented by the mitral E/e' ratio) [17].

## Limitations of ECHO-LVSWI

Controversy surrounds the ideal method to calculate the E/e' ratio for estimation of left ventricular filling pressures, with different authors and guidelines using medial, lateral or mean e' velocities in different settings [33–35]. Our institution has preferentially used the medial e'

**Table 5. Predictors of in-hospital mortality using multivariable logistic regression with stepwise backward variable selection to minimize the AIC value.**

| Variable | Adjusted OR | 95% CI | P value |
|---|---|---|---|
| *Demographics & comorbidities* | | | |
| Age (per 10 years) | 1.300 | 1.116–1.514 | 0.0008 |
| Female sex | 0.713 | 0.492–1.034 | 0.0741 |
| Charlson Comorbidity Index (per point) | 1.127 | 1.056–1.202 | 0.0003 |
| History of diabetes mellitus | 0.652 | 0.436–0.975 | 0.04 |
| Year of admission (per year) | 0.904 | 0.843–0.970 | 0.005 |
| *Admission diagnoses* | | | |
| Respiratory failure | 2.525 | 1.661–3.836 | <0.0001 |
| Sepsis | 2.106 | 1.294–3.429 | 0.0027 |
| *Severity of illness* | | | |
| APACHE score (per 10 points) | 1.223 | 1.096–1.364 | 0.0003 |
| Day 1 SOFA score (per point) | 0.890 | 0.813–0.974 | 0.01 |
| Admission Braden Skin Score (per point) | 0.900 | 0.844–0.960 | 0.0013 |
| *Shock severity* | | | |
| **SCAI shock stage (per stage)** | **1.243** | **1.024–1.508** | **0.03** |
| Inodilators in first 24 hours | 1.864 | 1.054–3.296 | 0.03 |
| *Procedures and therapies* | | | |
| Angiogram without PCI vs. no angiogram | 0.726 | 0.473–1.112 | 0.14 |
| PCI vs. no angiogram | 0.424 | 0.267–0.673 | 0.0003 |
| PCI vs. angiogram without PCI | 0.583 | 0.348–0.980 | 0.04 |
| pVAD or ECMO | 6.021 | 1.621–22.365 | 0.0073 |
| Dialysis | 2.005 | 0.826–4.866 | 0.12 |
| CRRT | 3.272 | 1.001–10.698 | 0.05 |
| *Cardiac arrest* | | | |
| VF CA vs. no CA | 3.036 | 1.795–5.135 | <0.0001 |
| Non-VF CA vs. no CA | 6.878 | 4.112–11.505 | <0.0001 |
| VF CA vs. non-VF CA | 2.266 | 1.246–4.121 | 0.007 |
| IHCA | 3.372 | 1.740–6.534 | 0.02 |
| *LVSWI (per 10 J/m2)* | *0.664* | *0.564–0.782* | *<0.0001* |

Data are displayed as adjusted odds ratio (OR) and 95% confidence interval (CI) values. The final model C-statistic value was 0.927 for discrimination of in-hospital mortality. Candidate variables which were not selected for the model included APACHE-IV predicted hospital mortality; race; invasive and noninvasive ventilator use; history of myocardial infarction, heart failure, chronic kidney disease, dialysis and stroke; peak VIS and NEE in first 24 hours; use of vasopressors in first 24 hours; IABP and PAC use; blood transfusion; CardShock score; LVEF; and admission diagnosis of heart failure or acute coronary syndrome.

velocity based on data showing superiority of the E/e' ratio using the medial e' for estimation of left ventricular filling pressures, and this is reflected in the data availability within our cohort [36]. However, using either the medial or lateral e' appears to have limitations under certain circumstances, and the use of the average or mean e' has been advocated [33–35]. In our analysis, we found that ECHO-LVSWI was minimally affected by using the medial, lateral, or mean e' velocity to estimate LVEDP, suggesting that this variable is minimally affected by the methodology used to calculate it and either lateral or mean e' velocity could be substituted. Furthermore, the presence of an elevated heart rate can produce fusion of the mitral E and A waves that poses a challenge to accurate assessment of the E/e' ratio during stress, as is typical of

patients with shock; notably, the ECHO-LVSWI had lower discrimination for patients with an elevated heart rate or greater shock severity [33,34].

We calculated ECHO-LVSWI automatically using data extracted from formal TTE reports in the medical record, as opposed to manual review of the primary TTE images. Calculating the ECHO-LVSWI by hand at bedside is time-consuming and prone to errors, including the potential for intra- and intra-observer variability, measurement inaccuracy and arithmetic errors during hand calculation. Accordingly, the time demand necessary to manually calculate the ECHO-LVSWI and the extent to which the inter- and intra-observer variability of manually calculated ECHO-LVSWI might influence its observed association with mortality remain uncertain. Therefore, it would be better to use an online calculator or ideally the ECHO-LVSWI could be calculated by the echocardiography imaging package or reporting software automatically. To facilitate use of ECHO-LVSWI, we have created an online calculator that can be used at the bedside (**S1 File**). Until automatic calculation of ECHO-LVSWI is incorporated into Doppler hemodynamic packages of bedside ultrasound machines or echocardiographic reporting software, we suggest that ECHO-LVSWI be calculated primarily using data obtained from a formal TTE to mitigate against these issues. Further research is necessary to determine when the potentially laborious calculation of ECHO-LVSWI is necessary, as opposed to the less-predictive but simpler SVI itself.

## Limitations of this study

This retrospective observational analysis cannot be used to determine cause-and-effect relationships, and the presence of residual confounding cannot be excluded as a mediator of the association between ECHO-LVSWI and outcomes. Only a relative minority of the entire CICU population had a TTE within one day of CICU admission containing complete data to calculate the ECHO-LVSWI, leading to potential selection bias and a lower-risk cohort than in our prior studies [3,6–11,17,24–28]. While this is typical of retrospective echocardiographic studies in critical care settings, we could not determine whether patient characteristics could have influenced which echocardiographic images were obtained and the quality of the data leading to further bias [10,17]. Without invasive hemodynamic data, we cannot be assured that the TTE accurately estimated the ECHO-LVSWI; we could not exclude the presence of poor Doppler signal alignment or other issues that could have affected the accuracy of our TTE measurements. Finally, we could not determine was vasoactive drugs patients were receiving at the time of TTE, which could have influenced the observed ECHO-LVSWI and its association with outcomes.

## Conclusions

ECHO-LVSWI was strongly and inversely associated with the risk of in-hospital mortality in CICU patients across the spectrum of cardiogenic shock severity, providing justification for its routine measurement in this population. At each SCAI shock stage, patients with a lower ECHO-LVSWI had higher in-hospital mortality, allowing ECHO-LVSWI to provide incremental risk stratification beyond the clinical assessment of shock severity. A lower ECHO-LVSWI identified high-risk patients in the group without overt shock, while a higher ECHO-LVSWI identified patients with hemodynamic instability or hypoperfusion who were at lower risk of adverse outcomes. Indeed, the presence of a low ECHO-LVSWI could reclassify patients at each SCAI shock into higher-risk subgroups with similar mortality to patients classified into a more severe SCAI shock stage. Our study emphasizes that hemodynamics (as assessed using Doppler TTE) can improve mortality risk stratification beyond the clinical definition of the SCAI shock stages. Future prospective studies are needed to better understand

the association between ECHO-LVSWI and outcomes in CS patients, and to determine whether interventions designed to improve the ECHO-LVSWI will translate into lower mortality in CICU patients.

## Supporting information

**S1 Fig.** Receiver-operator characteristic (ROC) curves demonstrating discrimination of in-hospital mortality by ECHO-LVSWI calculated using the medial (red), lateral (green) or mean (blue) e' velocity to estimate LVDEP for patients (n = 2896) with available data for both medial and lateral e' velocity. P values for comparison of AUC values were all >0.05 by De Long test.
(TIF)

**S2 Fig.** Receiver-operator characteristic (ROC) curves demonstrating discrimination of in-hospital mortality by ECHO-LVSWI (red), cardiac index (CI, green), LVEF (blue) and stroke volume index (SVI, orange). ECHO-LVSWI had a higher AUC value by the De Long test when compared with CI (p <0.0001), LVEF (p = 0.0002), or SVI (p = 0.06).
(TIF)

**S3 Fig.** Observed in-hospital mortality in patients grouped by quartiles of ECHO-LVSWI (red), CI (green), LVEF (blue) and SVI (orange). Median and interquartile range values defining the quartiles are as follows: ECHO LVSWI, 37.0 (21.0, 46.1) J/m2; CI, 2.8 (2.4, 3.3) L/min/ m2; LVEF, 51 (36, 61) %; SVI, 41 (33, 47) ml/m2.
(TIF)

**S1 Table. Measured and derived echocardiographic variables of interest.**
(DOCX)

**S2 Table. Formulas used to calculate echocardiographic hemodynamic parameters, using data from the time of the echocardiogram.**
(DOCX)

**S3 Table. Study definitions of hypotension, tachycardia, hypoperfusion, deterioration and refractory shock, as defined by Jentzer, et al. J Am Coll Cardiol 2019.**
(DOCX)

**S4 Table. Study definitions of Society for Cardiovascular Angiography and Intervention shock stages, as defined by Jentzer, et al. J Am Coll Cardiol 2019.**
(DOCX)

**S1 File. Automatic ECHO-LVSWI calculator based on 2-D and Doppler echocardiographic measurements.**
(XLSX)

## Author Contributions

**Conceptualization:** Jacob C. Jentzer, Brandon M. Wiley, Nandan S. Anavekar.

**Data curation:** Jacob C. Jentzer.

**Formal analysis:** Jacob C. Jentzer.

**Investigation:** Jacob C. Jentzer, Brandon M. Wiley, Nandan S. Anavekar.

**Methodology:** Jacob C. Jentzer, Brandon M. Wiley, Nandan S. Anavekar.

**Resources:** Jacob C. Jentzer.

**Validation:** Jacob C. Jentzer.

**Visualization:** Jacob C. Jentzer.

**Writing – original draft:** Jacob C. Jentzer.

**Writing – review & editing:** Jacob C. Jentzer, Brandon M. Wiley, Nandan S. Anavekar.

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
