## [Decision Letter · Decision Letter 0]

6 Oct 2021

PONE-D-21-26906Echocardiographic left ventricular stroke work index: An integrated noninvasive measure of shock severityPLOS ONE

Dear Dr. Jentzer,

Thank you for submitting your manuscript to PLOS ONE. After careful consideration, we feel that it has merit but does not fully meet PLOS ONE’s publication criteria as it currently stands. Therefore, we invite you to submit a revised version of the manuscript that addresses the points raised during the review process.

Please include the following items when submitting your revised manuscript:A rebuttal letter that responds to each point raised by the academic editor and reviewer(s). You should upload this letter as a separate file labeled 'Response to Reviewers'.A marked-up copy of your manuscript that highlights changes made to the original version. You should upload this as a separate file labeled 'Revised Manuscript with Track Changes'.An unmarked version of your revised paper without tracked changes. You should upload this as a separate file labeled 'Manuscript'.

We look forward to receiving your revised manuscript.

Kind regards,

Daniel A. Morris, M.D

Academic Editor

PLOS ONE

Journal Requirements:

Additional Editor Comments:

Thank you very much for submitting this excellent and large study to PlosOne. While the clinical relevance of the study is high, some pending `major limitations should be addressed in order to get adequate clinical applicability of the LVSWI.

Pending Limitations and Comments:

1) Concerns stated by the reviewers:

- The revisers have addressed important issues from this study, which should be mandatorily addressed in the revised version.

2) Uncertainty about the methodology of mitral E/e`:

- The mitral E/e ratio is an excellent parameter to estimate LV filling pressures, but however, its feasibility decreases significantly when the HR is > 90/min, which is a hallmark of patients in shock and/or with inotropic therapy. Hence, the authors should comprehensively discuss this issue and show how many patients had a fusion of E and A waves in both in the mitral inflow and in tissue Doppler velocities (i.e., e` and a´).

- Please describe with detail what type of mitral E/e ratio has been used in the study (i.e., average, septal, oder lateral mitral E/e` ratio).

3) Uncertainty of the clinical applicability of the LVSWI in patients with HR > 90/min:

- As it has been stated above, the low feasibility of the mitral E/e` in patients with tachycardia (a common and pathophysiological response in patients with shock) obligates to analyze alternative parameters when the mitral E/e` cannot be measured because of fusion of the E and A and e` and a´ waves. Hence, in order to increase the clinical relevance and applicability of the findings from this large and excellent study, the authors should further analyze and show the prognostic relevance of alternative echocardiographic parameters such as SVi, CI, and LVEF.

4) Lack of incremental value analyses concerning LVSWI:

- The authors should show the OR for intra-hospital mortality and the rate of intra-hospital mortality of the following parameters:

- LVSWI ≥ 33

- LVSWI < 33

- LVSWI < 20

- LVSWI < 10

- SVi ≥ 35

- SVi < 35

- SVi < 20

- SVi < 10

- CI ≥ 2,2

- CI < 2,2

- CI < 1,5

- CI < 1

- LVEF ≥ 50

- LVEF < 50

- LVEF < 40

- LVEF < 30

5) Incomplete description of main parameters in the main manuscript:

- Hemodynamic and echocardiographic variables were insufficiently described. Hence, the excellent supplemental tables 2 to 4 and supplemental figure 1 should be mandatorily included in the main manuscript.

Reviewers' comments:

Reviewer #1: 

This is an interesting article.

1-It would be ok if authors´ provide a figure(s) showing an example on how the left ventricular stroke work index (LVSWI) is measured by TTE.

2-Authors´mentioned that they used the medial (septal) e´velocity. This should be clarified and justified, particularly because e´septal is lower than e´lateral and thus the results may vary using either one or the other for calculations. The proper way is measuring both e´and average.

3- A point or limitation that I think may be mentioned is whether calculating the LVSWI is time-consuming enough to be performed in real time or not and also mentioning the possibilities of wrong calculations given the inherent error of each measurement. Who should best perform this calculations? a cardiologist? and intensivist? Who performed this calculations in your study?

Reviewer #2: 

In a study entitled “Echocardiographic left ventricular stroke work index: An integrated noninvasive measure of shock severity” Jentzer et al. in a retrospective analysis investigated a large group of patients admitted to CICU with cardiogenic shock. Authors found, that left ventricular stroke work index (LVSWI) derived noninvasively from echocardiography, can identify low-risk and high-risk patients at each level of clinical shock severity. LVSWI is a complex parameter combining systolic and diastolic function assessment.

The study is novel and interesting and has a practical aspect. The study is well written and is worth to be published.

Reviewer #3: 

The authors sought to evaluate the association of ECHO LVSWI with in-hospital mortality across SCAI shock stages and to determine whether early assessment of this hemodynamic variable could increase risk-stratification. This is a retrospective study that took place over an 8-year period between 2007 and 2015 and included 3,635 patients. The methodology is particularly rigorous and the results are very well presented. These results will undoubtedly have to be confirmed on prospective data and therapeutic solutions proposed in the context of subsequent work. No particular comment given the quality of the work done.

---

## [Author Response · Author response to Decision Letter 0]

25 Oct 2021

To:

Daniel A. Morris, M.D

Re: PONE-D-21-26906, Echocardiographic left ventricular stroke work index: An integrated noninvasive measure of shock severity

Dear Dr. Morris,

 We thank you for considering this revised manuscript for publication in PLoS One. We have addressed the comments of the Reviewers to the best of out ability, including several further analyses to satisfy their concerns. We recognize that the additive nature of this analysis building on our prior work may raise questions regarding its originality, but we believe that by synthesizing two previously-reported concepts, we have created a novel analysis that adds meaningfully to these prior works and stands on its own. We hope that the Reviewers and Editors will agree that this revised manuscript is substantially improved and that we have adequately justified the methodology reported. We hope that this revised manuscript is satisfactory for publication. Thank you for this opportunity.

Sincerely, 

 Jacob C. Jentzer, on behalf of the authors

Additional Editor Comments:

Thank you very much for submitting this excellent and large study to PlosOne. While the clinical relevance of the study is high, some pending `major limitations should be addressed in order to get adequate clinical applicability of the LVSWI.

Pending Limitations and Comments:

1) Concerns stated by the reviewers:

- The revisers have addressed important issues from this study, which should be mandatorily addressed in the revised version.

Authors’ Response: We have addressed the other Reviewer comments to the best of our ability. Please see detailed responses below.

2) Uncertainty about the methodology of mitral E/e`:

- The mitral E/e ratio is an excellent parameter to estimate LV filling pressures, but however, its feasibility decreases significantly when the HR is > 90/min, which is a hallmark of patients in shock and/or with inotropic therapy. Hence, the authors should comprehensively discuss this issue and show how many patients had a fusion of E and A waves in both in the mitral inflow and in tissue Doppler velocities (i.e., e` and a´).

- Please describe with detail what type of mitral E/e ratio has been used in the study (i.e., average, septal, oder lateral mitral E/e` ratio).

Authors’ Response: We appreciate these insightful comments regarding the limitations of the E/e’ ratio, which used the medial e’ in this case. We have added an entire paragraph regarding the challenges related to the use of E/e’ ratio as a valid estimate of LV filling pressures. To the Reviewer’s specific question regarding the effects of elevated HR on the performance of the ECHO-LVSWI for prediction of in-hospital mortality, we have performed additional secondary analyses. As the Reviewer has suspected, the LVSWI does not perform as well for mortality risk prediction in patients with HR >90. Regarding the question of E-A fusion, it is Mayo Clinic Echocardiography Laboratory policy not to report E velocities in patients with E-A fusion.

Manuscript excerpt:

Methods

“As per our prior study, we used the medial e’ velocity to calculate ECHO-LVSWI (Supplemental Figure 1); ECHO-LVSWI values were very similar when using either the lateral e’ velocity or mean e’ velocity (Pearson r correlation coefficients >0.99). Among patients with data for both medial and septal e’ velocities, there were no significant differences between the AUC values for discrimination of in-hospital mortality with ECHO-LVSWI calculated using the medial (0.756), lateral (0.751) or mean (0.754) e’ velocities (all p >0.05 by De Long test; Supplemental Figure 3). Per Mayo Clinic Echocardiography Laboratory policy, mitral E velocities were not reported for patients with E-A fusion.” 

Results

“The association between ECHO-LVSWI and in-hospital mortality was weaker for patients with a heart rate >90 beats/minute (unadjusted OR 0.585 per 10 J/m2 higher ECHO-LVSWI, 95% CI 0.442-0.776, AUC 0.645) compared with patients who had a slower heart rate (unadjusted OR 0.461 per 10 J/m2 higher ECHO-LVSWI, 95% CI 0.392-0.543, AUC 0.740).” 

Discussion

“Controversy surrounds the ideal method to calculate the E/e’ ratio for estimation of left ventricular filling pressures, with different authors and guidelines using medial, lateral or mean e’ velocities in different settings. Our institution has preferentially used the medial e’ velocity based on data showing superiority of the E/e’ ratio using the medial e’ for estimation of left ventricular filling pressures, and this is reflected in the data availability within our cohort. However, using either the medial or lateral e’ appears to have limitations under certain circumstances, and the use of the average or mean e’ has been advocated. In our analysis, we found that ECHO-LVSWI was minimally affected by using the medial, lateral, or mean e’ velocity to estimate LVEDP, suggesting that this variable is minimally affected by the methodology used to calculate it and either lateral or mean e’ velocity could be substituted. Furthermore, the presence of an elevated heart rate can produce fusion of the mitral E and A waves that poses a challenge to accurate assessment of the E/e’ ratio during stress, as is typical of patients with shock; notably, the ECHO-LVSWI had lower discrimination for patients with an elevated heart rate or greater shock severity.”

3) Uncertainty of the clinical applicability of the LVSWI in patients with HR > 90/min:

- As it has been stated above, the low feasibility of the mitral E/e` in patients with tachycardia (a common and pathophysiological response in patients with shock) obligates to analyze alternative parameters when the mitral E/e` cannot be measured because of fusion of the E and A and e` and a´ waves. Hence, in order to increase the clinical relevance and applicability of the findings from this large and excellent study, the authors should further analyze and show the prognostic relevance of alternative echocardiographic parameters such as SVi, CI, and LVEF.

Authors’ Response: We understand these comments and appreciate the concerns raised. We have highlighted the limitations of using ECHO-LVSWI in patients with elevated heart rate, as described above. The SVI, CI and LVEF were examined (albeit using single cut-offs rather than continuous analysis) in our prior manuscript examining echocardiographic variables across SCAI stages. As this analysis builds on the prior analysis, we are concerned that adding these data would be unnecessarily replicative. Regarding comparisons of ECHO-LVSWI to SVI, CI and LVEF, the AUC value for ECHO-LVSWI is higher than all of these other variables (p = 0.06 for SVI and p <0.05 for all others), see ROC curves below. For this reason, while we understand the desire to explore these other variables, we have done this in a prior analysis and believe that additional analysis of these variables is outside the scope of this manuscript.

Manuscript excerpt:

Results

“LVSWI had a higher AUC value (Supplemental Figure X) than LVEF (AUC 0.662, p <0.001), cardiac index (AUC 0.604, p <0.001) and SVI (AUC 0.702, p = 0.06).”

4) Lack of incremental value analyses concerning LVSWI:

- The authors should show the OR for intra-hospital mortality and the rate of intra-hospital mortality of the following parameters:

- LVSWI ≥ 33

- LVSWI < 33

- LVSWI < 20

- LVSWI < 10

- SVi ≥ 35

- SVi < 35

- SVi < 20

- SVi < 10

- CI ≥ 2,2

- CI < 2,2

- CI < 1,5

- CI < 1

- LVEF ≥ 50

- LVEF < 50

- LVEF < 40

- LVEF < 30

Authors’ Response: We respect the desire for additional data regarding echocardiographic hemodynamic alternatives to LVSWI for mortality risk stratification. As discussed above, this is to some extent outside the scope of this analysis and overlapping with our prior studies. Some of the cut-offs above were infrequently present in our cohort (i.e. CI <1, SVI <10, LVSWI <10) so instead of using these fixed cut-offs we have added a figure using quartiles of the variables in question to allow greater comparability between the variables (see below). This simple approach allows us to demonstrate the shape of these relationships and the spread between high-risk and low-risk patients identified using each of these variables in the same figure. This figure clearly shows how patients in the lowest LVSWI quartile have higher risk, and those in the highest LVSWI quartile have lower risk, when compared with quartiles of the other variables—this is the essence of risk stratification and nicely demonstrates the superiority of LVSWI. This is borne out by the OR values for each quartile versus the highest quartile (quartile 4), as shown below.

Manuscript excerpt:

Results

“When patients were grouped by quartiles of these variables, ECHO-LVSWI produced the greatest separation between high-risk and low-risk patients (Supplemental Figure Y and Supplemental Table 5).”

Quartile OR for LVEF OR for CI OR for SVI OR for LVSWI

1 3.91 (2.52-6.05) 2.62 (1.74-3.95) 7.21 (4.31-12.05) 15.83 (7.68-32.64)

2 1.99 (1.23-3.20) 1.12 (0.70-1.80) 3.48 (2.04-5.93) 6.97 (3.29-14.74)

3 1.17 (0.69-1.98) 1.25 (0.79-1.98) 2.04 (1.13-3.68) 3.84 (1.75-8.41)

4 (referent) 1.0 1.0 1.0 1.0

5) Incomplete description of main parameters in the main manuscript:

- Hemodynamic and echocardiographic variables were insufficiently described. Hence, the excellent supplemental tables 2 to 4 and supplemental figure 1 should be mandatorily included in the main manuscript.

Authors’ Response: We appreciate this comment. We have included the prior Supplemental Figure 1 as a main figure, as well as additional details about how ECHO-LVSWI is calculated. However, as the Tables are shared between this manuscript and other prior manuscripts using these same methods, we are uncertain whether it is appropriate to include them in the main document and will defer to the Editor’s final assessment in this regard.

Methods

“The mitral E/e’ ratio was used to estimate left ventricular end-diastolic pressure as LVEDP = 4.9 + 0.62 * mitral E/e’ ratio for calculation of ECHO-LVSWI using the formula ECHO-LVSWI = 0.0136 * stroke volume index (SVI) * (mean arterial pressure – LVEDP), as described by Choi, et al. (Supplemental Table 2).”

Reviewers' comments:

Authors’ Response: Regarding the question of dual publication, we have been transparent about the similarities and differences between this analysis and our previously published studies. This cohort includes a different group of patients, albeit overlapping partially with these prior analyses. The focus of this study is different insofar as we are exploring the interaction between the ECHO-LVSWI and the SCAI shock stages to evaluate how this integrated echocardiographic marker of cardiac systolic and diastolic performance complements a clinical assessment of shock severity for risk stratification. We believe that this analysis is a logical outgrowth from combining the concepts of the two prior studies yet has enough new and additive scientific data to stand on its own as a novel work. Finally, we believe that a study such as this one that includes a different patient group, has a different hypothesis, and performs distinct analyses is novel despite similarities to prior works, and should not be interpreted as a breach of research or publication ethics. 

Reviewer #1: 

This is an interesting article.

1-It would be ok if authors´ provide a figure(s) showing an example on how the left ventricular stroke work index (LVSWI) is measured by TTE.

Authors’ Response: We have added a Supplemental Figure 1 demonstrating this methodology, as shown below.

2-Authors´mentioned that they used the medial (septal) e´velocity. This should be clarified and justified, particularly because e´septal is lower than e´lateral and thus the results may vary using either one or the other for calculations. The proper way is measuring both e´and average.

Authors’ Response: We understand the Reviewer’s comment here. We used septal e’ in this analysis for a number of reasons. The septal/medial e’ was measured more consistently at our institution during the time period as evidenced by the greater availability of data for medial e’ velocities (lateral e’ velocities were missing for 739 patients); notably, this time period does predate contemporary guidelines and the measurements taken reflect the clinical practice at the time. This stems from prior research from our institution showing that “The correlations with the medial annulus TDI were consistently equivalent or better than the lateral annulus or the combinations of the medial and lateral annulus” for LV filling pressures, as described in Ommen, et al. Circulation 2000;102:1788-94. This was also the methodology we used in our prior published work on LVSWI (Jentzer, Circ Imaging 2020). Notably, the paper by Choi, et al. describing the echocardiographic estimation of LVSWI did not specify whether medial or lateral (or mean) e’ was used to estimate LVDEP. We have included a discussion of this important nuance. In addition, we re-calculated the LVSWI using either the lateral or mean/average E/e’ (in addition to medial/septal) and found that the correlations between these variables were >0.99 suggesting that the calculation of LVSWI was robust to the small differences arising from different methods of measuring E/e’. When we compared the AUC values for hospital mortality using these three methods of calculating LVSWI in patients with data for both e’ measurements, we found nearly identical AUC values that did not differ by the De Long test (see figure below); the LVSWI calculation using the medial e’ was borderline superior to that using the lateral e’ (p = 0.06). We feel that this justifies our use of medial/septal e’ velocity despite the other relevant considerations.

Methods

“As per our prior study, we used the medial e’ velocity to calculate ECHO-LVSWI (Supplemental Figure 1); ECHO-LVSWI values were very similar when using either the lateral e’ velocity or mean e’ velocity (Pearson r correlation coefficients >0.99). Among patients with data for both medial and septal e’ velocities, there were no significant differences between the AUC values for discrimination of in-hospital mortality with ECHO-LVSWI calculated using the medial (0.756), lateral (0.751) or mean (0.754) e’ velocities (all p >0.05 by De Long test; Supplemental Figure 2).”

Discussion

“Controversy surrounds the ideal method to calculate the E/e’ ratio for estimation of left ventricular filling pressures, with different authors and guidelines using medial, lateral or mean e’ velocities in different settings. Our institution has preferentially used the medial e’ velocity based on data showing superiority of the E/e’ ratio using the medial e’ for estimation of left ventricular filling pressures, and this is reflected in the data availability within our cohort. However, using either the medial or lateral e’ appears to have limitations under certain circumstances, and the use of the average or mean e’ has been advocated. In our analysis, we found that ECHO-LVSWI was minimally affected by using the medial, lateral, or mean e’ velocity to estimate LVEDP, suggesting that this variable is minimally affected by the methodology used to calculate it and either lateral or mean e’ velocity could be substituted.” 

3- A point or limitation that I think may be mentioned is whether calculating the LVSWI is time-consuming enough to be performed in real time or not and also mentioning the possibilities of wrong calculations given the inherent error of each measurement. Who should best perform this calculations? a cardiologist? and intensivist? Who performed this calculations in your study?

Authors’ Response: We thank the Reviewer for these questions. The ECHO-LVSWI calculations for this analysis were performed retrospectively using data from the database via an equation in the statical software package. We agree that hand calculation of ECHO-LVSWI at bedside is perhaps a bit onerous and prone to errors. If this could be incorporated into an app / online calculator or even better the echocardiographic calculation package that would be preferred.

Manuscript excerpt:

Discussion

“Calculating the ECHO-LVSWI by hand at bedside is tedious and prone to errors, so it would be better to use an online calculator or ideally the ECHO-LVSWI could be calculated by the echocardiography imaging package or reporting software automatically.” 

Reviewer #2: 

In a study entitled “Echocardiographic left ventricular stroke work index: An integrated noninvasive measure of shock severity” Jentzer et al. in a retrospective analysis investigated a large group of patients admitted to CICU with cardiogenic shock. Authors found, that left ventricular stroke work index (LVSWI) derived noninvasively from echocardiography, can identify low-risk and high-risk patients at each level of clinical shock severity. LVSWI is a complex parameter combining systolic and diastolic function assessment.

The study is novel and interesting and has a practical aspect. The study is well written and is worth to be published.

Authors’ Response: We thank the Reviewer for these constructive comments and appreciate the time taken to review our manuscript. 

Reviewer #3: 

The authors sought to evaluate the association of ECHO LVSWI with in-hospital mortality across SCAI shock stages and to determine whether early assessment of this hemodynamic variable could increase risk-stratification. This is a retrospective study that took place over an 8-year period between 2007 and 2015 and included 3,635 patients. The methodology is particularly rigorous and the results are very well presented. These results will undoubtedly have to be confirmed on prospective data and therapeutic solutions proposed in the context of subsequent work. No particular comment given the quality of the work done.

Authors’ Response: We thank the Reviewer for these constructive comments and appreciate the time taken to review our manuscript. We have emphasized the importance of prospective confirmation of our findings in the Conclusion.

Manuscript excerpt:

Conclusion

“Future prospective studies are needed to better understand the association between ECHO-LVSWI and outcomes in CS patients, and to determine whether interventions designed to improve the ECHO-LVSWI will translate into lower mortality in CICU patients.”

---

## [Editor Report · Decision Letter 1]

24 Nov 2021

PONE-D-21-26906R1Echocardiographic left ventricular stroke work index: An integrated noninvasive measure of shock severityPLOS ONE

Dear Dr. Jentzer,

Thank you for submitting your manuscript to PLOS ONE. After careful consideration, we feel that it has great merit but does not fully meet yet PLOS ONE’s publication criteria as it currently stands. Therefore, we invite you to submit a revised version of the manuscript that addresses the points raised during the review process.

Please include the following items when submitting your revised manuscript:A rebuttal letter that responds to each point raised by the academic editor . You should upload this letter as a separate file labeled 'Response to Reviewers'.A marked-up copy of your manuscript that highlights changes made to the original version. You should upload this as a separate file labeled 'Revised Manuscript with Track Changes'.An unmarked version of your revised paper without tracked changes. You should upload this as a separate file labeled 'Manuscript'.We look forward to receiving your revised manuscript.

Kind regards,

Daniel A. Morris, M.D

Academic Editor

PLOS ONE

Editor Comments:

I would like to congratulate again to the authors for this excellent large study as well as for the effort to address all suggestions of the editors and reviewers. The manuscript is almost ready for publication, there are only some minor pending limitations that should be mandatorily addressed to get the final version of this interesting study/manuscript.

Pending Limitations and Comments:

1) The time-consuming and the potential high inter- or intra-observer variability of the LVSWI:

- The time-consuming to calculate the LVSWI is a serious issue that should be comprehensively addressed and discussed.

- The potential high inter- or intra-observer variability of the LVSWI is another potential issue of this index.

- Accordingly, the authors should analyze in at least 20 patients the time-consuming of the LVSWI as well as the inter- und/or intra-observer variability of this index (i.e., the absolute inter- und/or intra-observer mean differences of the LVSWI in at least 20 patients).

- In addition, please provide fundaments and discuss why we should measure or use the LSWI instead LVEF, SVi, or CI. By the way, please discuss in which scenario we should add the LSWI to conventional systolic parameters (for instance, in those with septic shock and HR < 90 beats/min…?).

2) The potential incremental value of the LSWI over conventional LV systolic parameters:

- The supplemental figures X and Y and the supplemental table 5 are excellent and thus, it should be included in the main manuscript. By the way, please provide cutoff of the Youden index and the sensitivity and specificity of the cutoffs of the LVSWI, CI, LVEF, and SVi. In addition, please provide the values of the quartiles of the LVSWI, LVEF, CI, and SVi from the supplemental table 5.

---

## [Author Response · Author response to Decision Letter 1]

26 Nov 2021

Re: PONE-D-21-26906R1

“Echocardiographic left ventricular stroke work index: An integrated noninvasive measure of shock severity”

To:

Daniel A. Morris, M.D

Academic Editor

PLOS ONE

Dear Dr. Morris,

Thank you for considering this revised manuscript for publication in PLOS ONE. We have addressed the comments and requests to the best of our ability. We can clearly see the enthusiasm for a more extensive comparison of a variety of echocardiographic variables, but we want to emphasize that this study was designed with a rather limited scope—namely, to examine the ECHO-LVSWI in the context of the SCAI Shock classification. As such, extensive analyses of other echocardiographic variables outside of the structure of the SCAI Shock classification are not in line with our study hypothesis, and in some cases may impinge on our previously published works or distract from our main message. Therefore, in a few cases we have respectfully asked not to make the suggested changes even though we have tried to accommodate the suggested edits as much as possible. In addition, we have generated an online ECHO-LVSWI calculator to address concerns related to calculation time and arithmetic errors as a useful aid to clinicians who are interested in applying this variable. However, we are not able to hand-calculate ECHO-LVSWI from the primary echocardiographic images or using new patients based on the limits of our IRB approval. We hope that the Editor will understand these limitations and appreciate the efforts that we have made to address the remaining concerns and to ensure that this manuscript is a useful addition to the scientific literature. We appreciate your consideration.

Sincerely,

Jacob C. Jentzer, on behalf of the authors

Editor Comments:

I would like to congratulate again to the authors for this excellent large study as well as for the effort to address all suggestions of the editors and reviewers. The manuscript is almost ready for publication, there are only some minor pending limitations that should be mandatorily addressed to get the final version of this interesting study/manuscript.

Pending Limitations and Comments:

1) The time-consuming and the potential high inter- or intra-observer variability of the LVSWI:

- The time-consuming to calculate the LVSWI is a serious issue that should be comprehensively addressed and discussed.

- The potential high inter- or intra-observer variability of the LVSWI is another potential issue of this index.

Authors’ Response: We agree that the ECHO-LVSWI has important limitations in current practice related to the challenges in measuring and calculating ECHO-LVSWI at bedside. We have added a subheading “Limitations of ECHO-LVSWI” and amended the prior discussion to emphasize these points.

Manuscript excerpt:

“Calculating the ECHO-LVSWI by hand at bedside is time-consuming and prone to errors, including the potential for intra- and intra-observer variability, measurement inaccuracy and arithmetic errors during hand calculation. Therefore, it would be better to use an online calculator or ideally the ECHO-LVSWI could be calculated by the echocardiography imaging package or reporting software automatically. To facilitate use of ECHO-LVSWI, we have created an online calculator that can be used at the bedside (Supplemental X). Until automatic calculation of ECHO-LVSWI is incorporated into Doppler hemodynamic packages of bedside ultrasound machines or echocardiographic reporting software, we suggest that ECHO-LVSWI be calculated primarily using data obtained from a formal TTE to mitigate against these issues.”

- Accordingly, the authors should analyze in at least 20 patients the time-consuming of the LVSWI as well as the inter- und/or intra-observer variability of this index (i.e., the absolute inter- und/or intra-observer mean differences of the LVSWI in at least 20 patients).

Authors’ Response: We understand this request, but such an analysis extends beyond our IRB which covers only retrospective data gathering and analysis from the medical record, and not reviewing the echocardiographic images directly or collecting new patients prospectively. In this study, we calculated ECHO-LVSWI using data from formal TTE reports, and we advocate for using this approach in clinical practice at the current time. However, to try and assist providers in calculating the ECHO-LVSWI we have created formulas in an Excel spreadsheet as an online calculator, and this will help to facilitate the process and mitigate against arithmetic errors for end users. Using this calculator allows quick and accurate determination of ECHO-LVSWI and other hemodynamic calculations once the necessary data are measured (9 variables). We feel that providing this calculator for readers will be an important contribution, as we collaborate with imaging device companies to incorporate ECHO-LVSWI calculations into bedside ultrasound packages.

Manuscript excerpt:

“Calculating the ECHO-LVSWI by hand at bedside is time-consuming and prone to errors, including the potential for intra- and intra-observer variability, measurement inaccuracy and arithmetic errors during hand calculation. Therefore, it would be better to use an online calculator or ideally the ECHO-LVSWI could be calculated by the echocardiography imaging package or reporting software automatically. To facilitate use of ECHO-LVSWI, we have created an online calculator that can be used at the bedside (Supplemental File). Until automatic calculation of ECHO-LVSWI is incorporated into Doppler hemodynamic packages of bedside ultrasound machines or echocardiographic reporting software, we suggest that ECHO-LVSWI be calculated primarily using data obtained from a formal TTE to mitigate against these issues.”

- In addition, please provide fundaments and discuss why we should measure or use the LSWI instead LVEF, SVi, or CI. By the way, please discuss in which scenario we should add the LSWI to conventional systolic parameters (for instance, in those with septic shock and HR < 90 beats/min…?).

Authors’ Response: Our prior work has demonstrated that the LVEF and CI are inferior to SVI, which we have again shown using ROC analysis in this study (Supplemental Figure 3 as well as Supplemental Figure 4). We have likewise shown that LVSWI is superior to SVI, although this difference is of marginal significance (p = 0.06). We added these data to the prior revision. In addition, we have emphasized this point in the Discussion. At this time, it is premature to state that we know definitively which specific clinical scenarios call for use of ECHO-LVSWI as opposed to a different measurement, and our study was not designed to specifically answer this question. While we certainly understand the request to highlight the clinical value of the ECHO-LVSWI in this way, we do not want to make statements that are not supported by our analyses. This study was designed to establish the relevance of ECHO-LVSWI in the context of the SCAI shock classification for mortality risk stratification. Further research is clearly needed to extend the clinical utility of ECHO-LVSWI beyond risk stratification and into clinical decision-making.

Manuscript excerpt:

Results

“LVSWI had a higher AUC value (Supplemental Figure 3) than LVEF (AUC 0.662, p <0.001), cardiac index (AUC 0.604, p <0.001) and SVI (AUC 0.702, p = 0.06). When patients were grouped by quartiles of these variables, ECHO-LVSWI produced the greatest separation between high-risk and low-risk patients (Supplemental Figure 4 and Supplemental Table 5).”

Discussion

“We have demonstrated that ECHO-LVSWI has higher discrimination for in-hospital mortality than LVEF, SVI and CI when analyzed as continuous variables, providing risk stratification by separating high-risk and low-risk CICU patients.” 

“ECHO-LVSWI may prove to be a superior non-invasive parameter because it integrates diastolic function assessment, providing deeper insights into overall myocardial function that translates into improved mortality risk stratification.” 

“Further research is necessary to determine when the potentially laborious calculation of ECHO-LVSWI is necessary, as opposed to the less-predictive but simpler SVI itself.”

2) The potential incremental value of the LSWI over conventional LV systolic parameters:

- The supplemental figures X and Y and the supplemental table 5 are excellent and thus, it should be included in the main manuscript. By the way, please provide cutoff of the Youden index and the sensitivity and specificity of the cutoffs of the LVSWI, CI, LVEF, and SVi. In addition, please provide the values of the quartiles of the LVSWI, LVEF, CI, and SVi from the supplemental table 5.

Authors’ Response: We perceive the Reviewer’s strong interest in these comparisons between echocardiographic variables, yet we want to emphasize that this study was not designed to simply validate the ECHO-LVSWI in comparison to other TTE variables, but rather to examine the ECHO-LVSWI in the context of shock severity defined by SCAI shock stage (and not as a standalone risk stratification tool). Our prior work has examined LVSWI as a standalone risk stratification tool and has explored LVEF, SVI and CI in the context of SCAI shock stage. We worry that these specific figures and tables might distract from the primary message. As such, we feel that these supplemental figures (which are not related to SCAI shock stage) are tangential to the main hypothesis and respectfully request that they remain supplemental for this reason; if the Editor feels strongly, we can include them as main figures. As requested, we have added the median and IQR values (which define the quartiles) to the figure legends and have added a Supplemental Table 6 with the requested diagnostic performance of the echocardiographic variables.

Manuscript excerpt:

Results

“At the optimal cut-off, LVSWI had the highest combined sensitivity and specificity (Supplemental Table 6).”

---

## [Editor Report · Decision Letter 2]

1 Dec 2021

PONE-D-21-26906R2Echocardiographic left ventricular stroke work index: An integrated noninvasive measure of shock severityPLOS ONE

Dear Dr. Jentzer,

Thank you for submitting your manuscript to PLOS ONE. After careful consideration, we feel that it has merit but does not fully meet PLOS ONE’s publication criteria as it currently stands. Therefore, we invite you to submit a revised version of the manuscript that addresses the points raised during the review process.

Please include the following items when submitting your revised manuscript:A rebuttal letter that responds to each point raised by the academic editor and reviewer(s). You should upload this letter as a separate file labeled 'Response to Reviewers'.A marked-up copy of your manuscript that highlights changes made to the original version. You should upload this as a separate file labeled 'Revised Manuscript with Track Changes'.An unmarked version of your revised paper without tracked changes. You should upload this as a separate file labeled 'Manuscript'.We look forward to receiving your revised manuscript.

Kind regards,

Daniel A. Morris, M.D

Academic Editor

PLOS ONE

Additional Editor Comments:

Thank you very much again for the efforts to address the pending limitations from this study. While the revised version has improved, it remains some limitations that should be mandatorily addressed to get priority for publishing this study in PlosOne.

Pending Limitations:

1) Please highlight in the limitations section that “the echocardiographic data was merely obtained from medical records. Accordingly, it remains uncertain what would be the time-consuming and the inter- and intra-observer variability of the LVSWI”.

2) The following figures and tables should be completed and mandatorily incorporated into the main manuscript to improve the presentation of this study:

- Supplemental Figures 1a and 1b.

- Supplemental Table 5 (please add the values of the quartiles in this table).

- Supplemental Table 6 (please delete the file “Youden’s J index @ cut-off” in this table).

---

## [Author Response · Author response to Decision Letter 2]

1 Dec 2021

Re: PONE-D-21-26906R2

“Echocardiographic left ventricular stroke work index: An integrated noninvasive measure of shock severity”

To:

Daniel A. Morris, M.D

Academic Editor

PLOS ONE

Dear Dr. Morris,

Thank you for considering this revised manuscript for publication in PLOS ONE. We have addressed all the comments and requests as instructed. We hope that you will find this revised manuscript to be acceptable for publication.

Sincerely,

Jacob C. Jentzer, on behalf of the authors

Additional Editor Comments:

Thank you very much again for the efforts to address the pending limitations from this study. While the revised version has improved, it remains some limitations that should be mandatorily addressed to get priority for publishing this study in PlosOne.

Pending Limitations:

1) Please highlight in the limitations section that “the echocardiographic data was merely obtained from medical records. Accordingly, it remains uncertain what would be the time-consuming and the inter- and intra-observer variability of the LVSWI”.

Authors’ Response: This suggested addition has been made, although we have used our own wording.

Manuscript excerpt:

Discussion

“We calculated ECHO-LVSWI automatically using data extracted from formal TTE reports in the medical record, as opposed to manual review of the primary TTE images. Calculating the ECHO-LVSWI by hand at bedside is time-consuming and prone to errors, including the potential for intra- and intra-observer variability, measurement inaccuracy and arithmetic errors during hand calculation. Accordingly, the time demand necessary to manually calculate the ECHO-LVSWI and the extent to which the inter- and intra-observer variability of manually calculated ECHO-LVSWI might influence its observed association with mortality remain uncertain.” 

2) The following figures and tables should be completed and mandatorily incorporated into the main manuscript to improve the presentation of this study:

- Supplemental Figures 1a and 1b.

- Supplemental Table 5 (please add the values of the quartiles in this table).

- Supplemental Table 6 (please delete the file “Youden’s J index @ cut-off” in this table).

Authors’ Response: These suggested additions have been made, and the tables and figures have been renumbered.

---

## [Editor Report · Decision Letter 3]

16 Dec 2021

Echocardiographic left ventricular stroke work index: An integrated noninvasive measure of shock severity

PONE-D-21-26906R3

Dear Dr, Jentzer,

We’re pleased to inform you that your manuscript has been judged scientifically suitable for publication and will be formally accepted for publication once it meets all outstanding technical requirements.

Kind regards,

Daniel A. Morris, M.D

Academic Editor

PLOS ONE

---

## [Editor Report · Acceptance letter]

1 Mar 2022

PONE-D-21-26906R3 

Echocardiographic left ventricular stroke work index: An integrated noninvasive measure of shock severity 

Dear Dr. Jentzer:

I'm pleased to inform you that your manuscript has been deemed suitable for publication in PLOS ONE. Congratulations! Your manuscript is now with our production department. 

Kind regards, 

on behalf of

Dr. Daniel A. Morris 

Academic Editor

PLOS ONE